# Case of Prenatal Diagnosis of a Fetal Pulmonary Arteriovenous Malformation at Term

**DOI:** 10.3390/diagnostics14090876

**Published:** 2024-04-24

**Authors:** Kristina Tchatcheva, Rumen Marinov

**Affiliations:** 1Nadezhda Women’s Health Hospital, 1373 Sofia, Bulgaria; 2National Cardiology Hospital, 1309 Sofia, Bulgaria

**Keywords:** arteriovenous malformation, prenatal diagnosis, 3D-CT reconstruction, surgical treatment

## Abstract

In our case, the malformation was diagnosed prenatally at 40 weeks of gestation, and at the age of 14 days, the malformation was removed combined with a segmentectomy of the sixth segment of the left lung. Preoperative diagnostics focus on 3D-CT reconstruction and detailing of the anatomical variations of all arterial and venous vessels, as evident from our case. Treatment includes surgical removal or a minimally invasive interventional approach through the embolization of the vessel afferent to the malformation. After the operation, the child was discharged on the 30th day after birth in good condition and is developing normally. Early operative intervention is of great importance for the favorable outcome of the condition. In our case, this was hypoxemia with a saturation of 70-75%. The rare and often missed prenatal diagnosis of fetal AV malformation is significant for the adequate postnatal treatment and development of affected children.

The pulmonary arteriovenous malformations (PAVM) are abnormal vascular structures that link a pulmonary vein and artery, constituting a bypass of the pulmonary capillary beds. Prenatal diagnosis of a fetal PAVM is very rare and often misplaced. Diagnosis is hard and is often placed in late childhood due to unexplained cyanosis and/or other complications—most commonly hemoptysis. It is believed that most cases of PAVM are congenital. They often appear with congenital hemorrhagic telangiectasia [1] but can also be idiopathic, connected with trauma or inflammation. During fetal life PAVM can lead to serious cardiac dysfunction, cardiac insufficiency, and fetal death. During childhood, they can lead to hypoxia, paradoxical embolism, and cerebral infarctions. The literature describes a few cases of such malformations. Some of them were diagnosed during the prenatal period, the earliest being during the early second trimester [1]. The few described cases of fetal PAVM had different outcomes—intrauterine fetal loss and favorable neonatal outcomes [2,3,4] (Figure 1, Figure 2, Figure 3, Figure 4, Figure 5 and Figure 6).

## Figures and Tables

**Figure 1 diagnostics-14-00876-f001:**
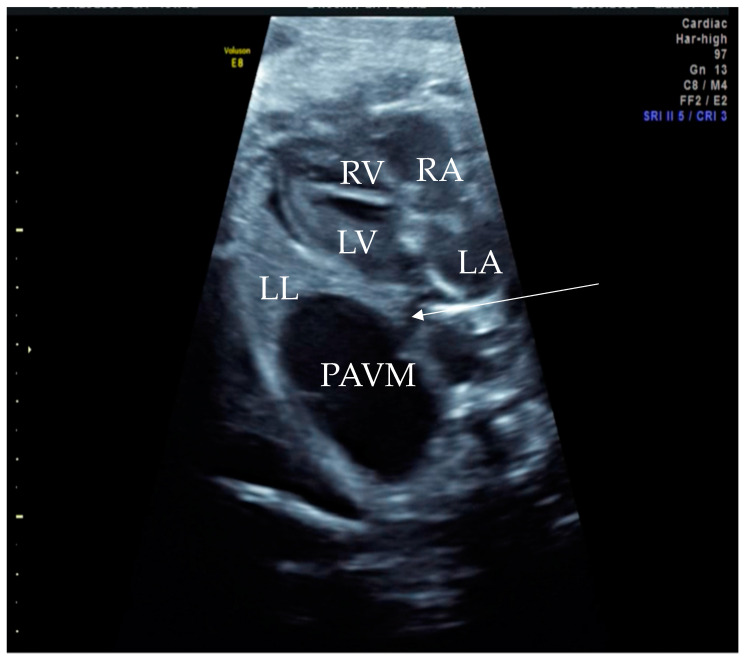
Prenatal B-Mode transabdominal ultrasound picture with cystic structure on the level of four chamber view with a clear linkage of the structure to the left atrium. RV—right ventricle, LV—left ventricle, RA—right atrium, LA—left atrium, LL—left lung, PAVM—pulmonary arteriovenous malformation, arrow—showing the connecting vessel to the left atrium and the PAVM. Actually, this finding led to the suspicion of PAVM. We needed to prove it with color Doppler, as shown in the next picture.

**Figure 2 diagnostics-14-00876-f002:**
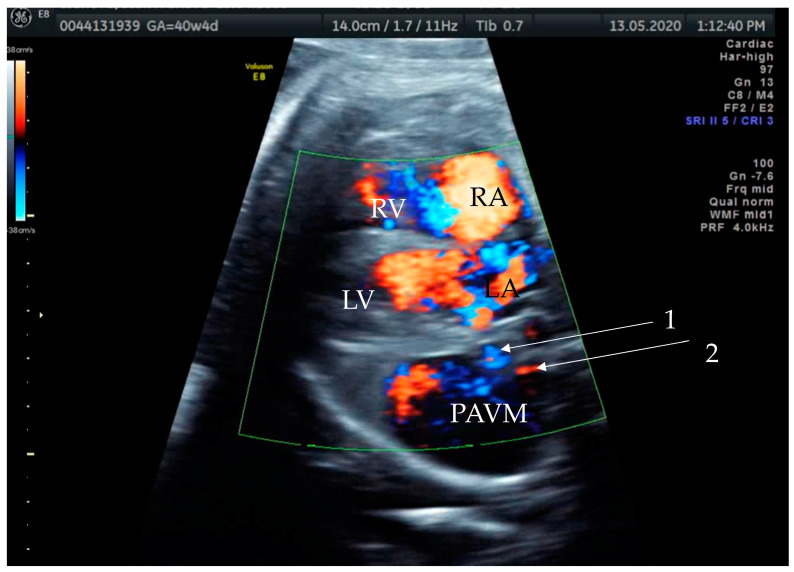
Prenatal transabdominal color Doppler mode ultrasound picture with visible bidirectional flow in the cystic structure. Arrow 1—venous part of the feeding vessels with connection to the left atrium, Arrow 2—arterial part of the connection to the left main branch of the pulmonal artery, RV—right ventricle, LV—left ventricle, RA—right atrium, LA—left atrium, and PAVM—pulmonary arteriovenous malformation. Due to the gestational age, it was difficult to distinguish the two feeding vessels in one picture, as shown in the last prenatal picture (Figure 3).

**Figure 3 diagnostics-14-00876-f003:**
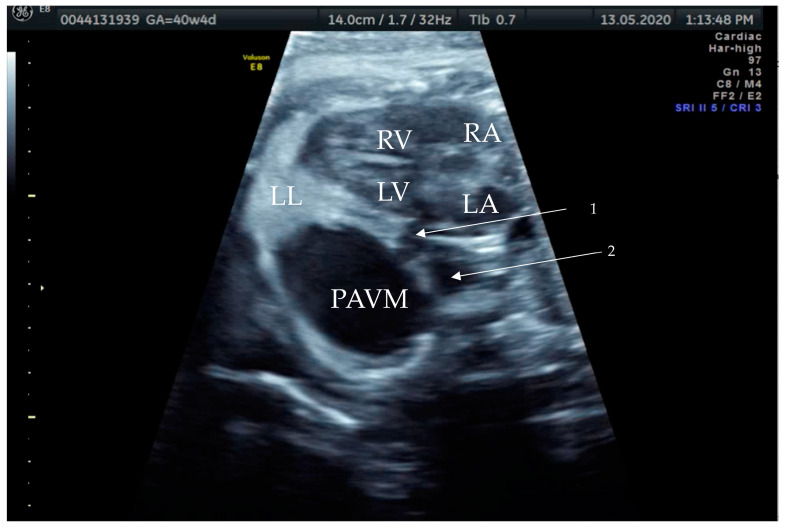
Prenatal B-Mode transabdominal ultrasound picture of the PAVM with a clear linkage of the structure to the left atrium and a visible second vessel, more dorsal and cranial, appearing to be the arterial part of the feeding vessels connecting to the left main branch of the pulmonary artery. RV—right ventricle, LV—left ventricle, RA—right atrium, LA—left atrium, LL—left lung, PAVM—pulmonary arteriovenous malformation, arrow 1—showing the venous vessel connecting the left atrium and the PAVM, and arrow 2—showing the arterial vessel connecting the PAVM.

**Figure 4 diagnostics-14-00876-f004:**
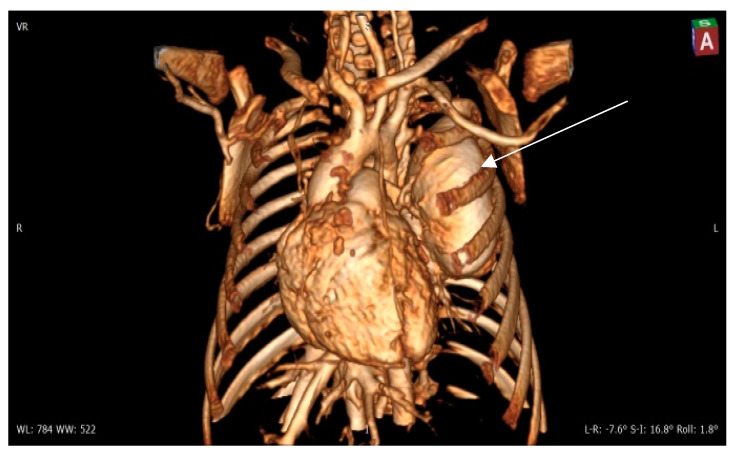
Contrast 3D-CT reconstruction of the structure in the left lung—arrow directed to the PAVM. A rounded formation is present in the left lung constituting the “external manifestation” of the arteriovenous fistula.

**Figure 5 diagnostics-14-00876-f005:**
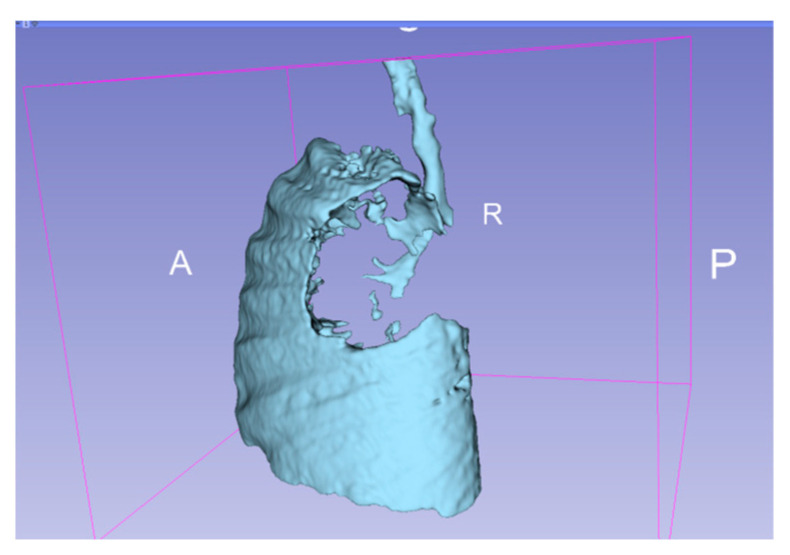
3D representation of the affected areas of the lung. A—anterior, P—posterior, R—right.

**Figure 6 diagnostics-14-00876-f006:**
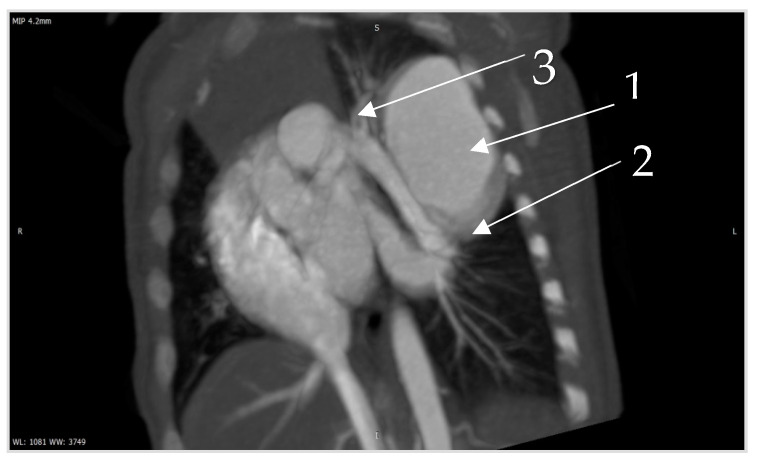
Contrast CT—native image of the branches of the pulmonary artery for three lower and two upper segments. Preoperative diagnostics focus on 3D-CT reconstruction and detailing of the anatomical variations of all arterial and venous vessels, as evident from our case. Contrast CT (Figure 4, Figure 5 and Figure 6) shows communication between the left lower branch of the pulmonary artery—an aneurysmal formation—arrow 1 in Figure 6, measuring 35 × 30 × 25 mm, and drainage into a wide vessel with an irregular shape and through the left inferior pulmonary vein—arrow 2 in Figure 6 into the left atrium—arrow 3 in Figure 6 (independent branching of the subsegmental branches of the LPA for three lower segments and two upper). The inlet and outlet vessels of the AV malformation are shown. The bronchography shows a lack of lung structures in the location of the aneurysmal formation. There are also no signs of pressure and/or hypoplasia in the rest of the left lung.

## Data Availability

Not applicable.

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
