# Peer review of "Case of Prenatal Diagnosis of a Fetal Pulmonary Arteriovenous Malformation at Term"

_diagnostics, 2024, doi:10.3390/diagnostics14090876_

Round 1

Reviewer 1 Report

Comments and Suggestions for Authors

The authors present an interesting case of a pulmonary AVM diagnosed in utero.  The images presented are very good.  I think it would add more to the manuscript if they were able to provide a pathology specimen.

Comments on the Quality of English Language

There is moderate editing required and some of the meaning is lost due to grammatical errors.  

Author Response

Dear colleagues, unfortunately in the pathohistological laboratory , there is no preserved material of the vascular structure after the operation. Our goal  is to show the structure as an opportunity for prenatal diagnosis.

Reviewer 2 Report

Comments and Suggestions for Authors

Dear Authors,

This is a very interesting case of AV malformation in the lung. Was this the first presentation in your clinic? Was the baby anemic? Please include a brief differential diagnosis of prenatal lung masses. For those practicing fetal medicine, what are the prenatal findings that can suggest a PAVM? Can we have a conclusion?

What was the final outcome of the baby?

Comments on the Quality of English Language

Good English, just minor modifications

Author Response

Thank you very much for taking the time to review this manuscript. Please see the attachment.

Reviewer 3 Report

Comments and Suggestions for Authors

line 18: specify  the indications for early surgery

line 19-20: describe the hemodynamic compensation assessed by echocardiogram

Was the degree of right-left shunt assessed? how? Was contrast echocardiogram performed?

Figure 6 is not clearly explained and also includes 5 . furthermore, it is not clear what the last few lines refer to (still figure 5? from number 6 cannot be deduced )

line 25- in the conclusions it could be explained more clearly whether early diagnosis improves the child's long-term development (pulmonary?), prognosis , symptoms, etc.

Author Response

Thank you very much for taking the time to review this manuscript. Please find the detailed responses below.

Reviewer 4 Report

Comments and Suggestions for Authors

The authors provided some interesting images, but for better understanding some key information needs to be included.

Description of the patient's condition, PAVM diagnosis time, weight, clinical status (blood oxygen saturation, Apgar scores…) Authors also mentioned ‘outcome of the condition’, the clinical follow-up examinations are needed if applicable.

Detailed description of echocardiography.

The pulsed-wave Doppler images need to be shown.

It would be better to provide motion images.

Author Response

(The authors gave the same response as above.)

Round 2

Reviewer 4 Report

Comments and Suggestions for Authors

The authors made some improvements, but demonstrating the pulsed-wave Doppler findings is crucial.